# Graphene as Nanocarrier for Gold(I)-Monocarbene Complexes: Strength and Nature of Physisorption

**DOI:** 10.3390/molecules28093941

**Published:** 2023-05-07

**Authors:** Cahit Orek, Massimiliano Bartolomei, Cecilia Coletti, Niyazi Bulut

**Affiliations:** 1Department of Physics, Faculty of Science, Firat University, Elazig 23119, Turkey; cahitorek@gmail.com; 2Instituto de Fisica Fundamental, Consejo Superior de Investigaciones Cientificas (IFF-CSIC), Serrano 123, 28006 Madrid, Spain; maxbart@iff.csic.es; 3Dipartimento di Farmacia, Università degli Studi “G. d’Annunzio” Chieti-Pescara, Via dei Vestini, 66100 Chieti, Italy

**Keywords:** gold complexes, graphene, drug nanocarriers, metal based anticancer compounds, DFT

## Abstract

Gold(I) metal complexes are finding increasing applications as therapeutic agents against a variety of diseases. As their potential use as effective metallodrugs is continuously confirmed, the issue of their administration, distribution and delivery to desired biological targets emerges. Graphene and its derivatives possess attractive properties in terms of high affinity and low toxicity, suggesting that they can efficaciously be used as drug nanocarriers. In the present study, we computationally address the adsorption of a gold(I) N-heterocyclic monocarbene, namely, IMeAuCl (where IMe = 1,3-dimethylimidazol-2-ylidene), on graphene. The Au(I) N-heterocyclic carbene family has indeed shown promising anticancer activity and the N-heterocyclic ring could easily interact with planar graphene nanostructures. By means of high-level electronic structure approaches, we investigated the strength and nature of the involved interaction using small graphene prototypes, which allow us to benchmark the best-performing DFT functionals as well as assess the role of the different contributions to total interaction energies. Moreover, realistic adsorption enthalpies and free energy values are obtained by exploiting the optimal DFT method to describe the drug adsorption on larger graphene models. Such values (ΔH_ads_ = −18.4 kcal/mol and ΔG_ads_= −7.20 kcal/mol for the largest C_150_H_30_ model) indicate a very favorable adsorption, mainly arising from the dispersion component of the interaction, with the electrostatic attraction also playing a non-negligible role.

## 1. Introduction

Gold(I) N-heterocyclic carbenes (NHCs) have long been known for their applications in homogeneous catalysis [1], and, more recently, are widely investigated in medicine [2] as therapeutical agents as antibacterial [3], antiarthritic [4] and antitumoral [5,6,7,8,9,10] compounds. In the latter case, their mechanism of action seems to be specifically targeted to proteins [11,12] mainly binding to cysteine and selenocysteine residues [13,14,15], as is the case for other coinage metal-based N-heterocyclic carbenes [16]. Indeed, gold(I) shows a high affinity for selenol and thiol groups making enzymes such as thioredoxin reductase and glutathione reductase their preferred targets [9,17,18,19,20,21], inhibiting their corresponding activities, and thus leading to mitochondrial damage and cell apoptosis. In recent years much work has been devoted to clarifying the action mode of gold(I) mono- and bis-carbenes both experimentally [11,19,20,21] and theoretically [10,12,14,15]. Monocarbenes have the general formula (NHC)AuX, where X- is a labile ligand, usually a halide ion, which can be easily substituted by thiols or selonols in physiological conditions; biscarbenes, ionic compounds containing the [M(NHC)_2_]^+^ ion, can instead only bind their target after the loss of a carbene ligand.

Though much work has been carried out to evaluate the effectiveness of such compounds and to elucidate the reasons behind their binding mechanism, less attention has been paid to investigations concerning their administration and distribution properties. Molecular nanocarriers have found increasing applications for specific drug delivery, particularly important in the case of cancer-related therapy [22,23], where specificity is one of the key issues to overcome or reduce the often-severe side effects associated with chemotherapy. Moreover, the nanocarriers’ solubility and lipo/hydrophilicity properties can have a strong impact on the facility of drug administration; therefore, their analysis contributes to the choice of the most effective drug delivery system. Since their discovery, graphene layers and quantum dots, and their derivatives, have been considered as promising nanosystems for specific drug delivery [23,24,25,26]. As a matter of fact, graphene-based materials enjoy a wealth of properties which can be easily exploited for drug administration, load and delivery. They show a high surface/volume ratio, allowing for remarkable drug loading. The possibility of being decorated with different functional groups, as is the case for graphene oxide, allows the modulation of the hydrophilicity of the support and thus the solubility in aqueous environments, as well as the charge/discharge conditions depending on the cellular chemical environment, and many other physico-chemical properties [27]. Therefore, the use of graphene, graphene quantum dots and different types of graphene oxide has been explored in conjunction with different antitumoral compounds and often demonstrated to enhance the cellular uptake in cancer tissues and thus the cytotoxic activity [28,29,30,31,32]. This is the case for cisplatin, cis-[PtCl_2_(NH_3_)_2_], one of the first and still used anticancer molecules, for which, when loaded on graphene oxide platelets, remarkable pH-dependent uptake/release effects were found [32]: alkaline pH values significantly enhance the amount of cisplatin charged on GO supports, while acidic conditions increase its release, a desirable mechanism when considering the hypoxic environment of tumoral cells. 

Recent theoretical studies [33,34,35] have addressed the molecular adsorption mechanism of cisplatin onto graphene and graphene oxide supports, to elucidate the viability of physisorption and chemisorption processes. These studies do not find a counterpart when other metal-based molecules, such as the promising gold(I) complexes, are considered. In the present paper, we therefore computationally investigate the possibility and nature of the adsorption of gold(I) NHC complexes onto graphene sheets. The simple, but widely studied, gold(I) N-heterocyclic monocarbene, IMeAuCl (where IMe = 1,3-dimethylimidazol-2-ylidene), is taken as the prototype of the Au(I) N-heterocyclic carbene family (Figure 1), whereas different graphene molecular models, with sizes ranging from C_37_H_16_ to C_150_H_30_, were considered. The smallest one allows us to select the best computationally affordable level of theory, when benchmarked against highly accurate post-HF methods, and to study the physical components of the interaction between IMeAuCl and graphene in detail. The increasing size of the larger models allows us to consider the convergence of the calculation with respect to more realistic representations of the graphene sheet and to accurately calculate the thermodynamics of the adsorption process. 

The article is organized as follows. The obtained results are described, analyzed and discussed in Section 2, where enthalpy and free energy for the adsorption of IMeAuCl onto graphene are also given. Section 3 describes the computational methodologies used to benchmark the DFT functionals to be selected for the calculations involving the largest graphene prototypes and to analyze the physical contributions to the total interaction energy. Conclusions and remarks are given in Section 4 together with prospective future work.

## 2. Results and Discussion

For benchmarking purposes, we have used C_37_H_16_ (Figure 1) as an initial model for the graphene plane: it is a planar polycyclic aromatic hydrocarbon (PAH) small enough to be handled using high-level electronic structure theories necessary to recover the involved weak intermolecular interactions to investigate the physical nature of the adsorption and, at the same time, large enough to allow for the determination of the interaction with IMeAuCl with a reduced contribution of the in-plane peripheral hydrogen atoms. The effect of larger, more realistic graphene models will then be evaluated at a selected computationally more affordable DFT level of theory.

We investigated several different initial mutual geometries for the C_37_H_16_–IMeAuCl complex, mostly with the graphene plane and the IMeAuCl five-member ring arranged in a parallel fashion, since this configuration is expected to lead to the most favorable interactions. Minimization invariably led to the minimum depicted in Figure 1, showing that the five-member ring lies parallel to the graphene plane and the gold atom interacts with the π electron cloud of the graphene model. The energy profiles, corresponding to the parallel approach of rigid IMeAuCl to C_37_H_16_ according to such configuration, are reported in Figure 2 as a function of R, the distance between IMeAuCl and C_37_H_16_ centers of mass, ranging from 2.5 to 8.0 Å. They were calculated at different levels of theory: the reference DFT-SAPT/CBS and MP2C/CBS approaches as well as the PBE-D3(BJ), M062X, and B3LYP-D3(BJ) functionals, as described in the Computational Methods section. All energy values are BSSE corrected and can be found in Appendix A.

At all levels of theory, the minimum configuration is found at R ≈ 3.3 Å. The two reference profiles, DFT-SAPT/CBS and MP2C/CBS, provide a lower and an upper limit of the involved interactions, with a minimum and maximum interaction energy of ca. −770 meV and −680 meV, respectively. PBE-D3(BJ) results are very close to the DFT-SAPT reference and only slightly more attractive at long range. For M062X, though the well depth is approximately the same, the equilibrium distance R is sensibly shorter, and the long-range potential is much less attractive, which could be due to documented faults of the M062X functional in the asymptotic regions [36]. B3LYP-D3(BJ) results present the largest discrepancies: the well is roughly 200 meV deeper than DFT-SAPT, PBE-D3(BJ) and M062X, and 100 meV deeper than MP2C, and is found at shorter R values. These results are in agreement with previous studies on the interaction of graphene with metal-based complexes [33,34,37] and show that the PBE-D3(BJ) functional lying in between MP2C/CBS and DFT-SAPT/CBS, only slightly overestimating the latter, can be safely used to carry out computationally more demanding characterizations, such as those involving more realistic graphene models.

The interaction energy decomposition into its physically significant contributions has been examined at the DFT-SAPT level of theory, allowing us to characterize the nature of the adsorption of IMeAuCl onto the C_37_H_16_ prototype. The results are shown in Figure 3, and the corresponding energies at selected R values are displayed in Appendix A. Table 1 reports the DFT/SAPT contributions for the minimum geometry calculated at the PBE-D3(BJ)/6-311++G(2d,2p)-SDD level of theory (see the following). The main responsibility of the binding on graphene is the dispersion contribution, as could be expected due to the non-polar nature of graphene and the planar structure of the adsorbate. The electrostatic component amounts only to about one-quarter of the total attraction energy at a minimum. The latter component becomes more important at lower interaction distances when the repulsive exchange tends to predominate. Dispersion interactions start to be effective at very long range where they are the only contribution to the total interaction. The large dispersion contribution (more than 60% of the overall attraction energy) suggests that the nature of the interaction resembles π-π stacking for whose description the PBE-D3(BJ) functional was found to work very well, outperforming B3LYP-D3(BJ) [37].

The optimization of the adsorption geometry was therefore carried out at the PBE-D3(BJ)/6-311++G(2d,2p)-SDD level of theory (as well as at B3LYP-D3(BJ) and M062X) by considering either rigid or flexible C_37_H_16_. A preliminary minimization of the isolated IMeAuCl structure shows that the calculated values accurately match the crystallographic data [39], with Au–C1 and C–N bond distances within 0.008 and Cl–Au–C1 angles within 1.1, indicating that the PBE-D3(BJ) functional provides a good description of this molecular structure (Table 2). In the same table, the geometry of IMeAuCl adsorbed on C_37_H_16_ at the equilibrium distance is also reported and shown to be substantially unmodified, with only a slight elongation (0.01 Å) of the Au-Cl bond distance.

The effect of C_37_H_16_ geometry relaxation upon IMeAuCl adsorption has also been investigated: the atom-by-atom superimposition of the structures for a fully relaxed (red) over a rigid structure (black) is shown in Figure 4. A very slight deviation from planarity is observed, with the peripheral carbon and hydrogen atoms pointing to IMeAuCl to maximize the interaction. BSSE-corrected interaction energies were determined in both cases and are reported in Table 3. In the same table, the energy of the minimum calculated with PBE-D3(BJ), B3LYP-D3(BJ) and M062X functionals with the 6-311++G(2d,2p)+SDD basis set and single point energies calculated on the PBE-D3(BJ)/6-311++G(2d,2p)+SDD geometries at the reference DFT-SAPT/CBS and MP2C/CBS levels of theory are also shown. As observed for the energy profiles in Figure 2 the E_int_ values calculated at PBE-D3(BJ)/6-311++G(2d,2p)+SDD lie in between those obtained at the MP2C and DFT-SAPT level, being only slightly lower (about 20–30 meV) than the latter. Conversely, B3LYP-D3(BJ) largely overestimates the binding energy, and the minimum interaction distance R is shorter than PBE-D3(BJ) (R = 3.35 Å for PBE-D3(BJ), R = 3.28 Å for B3LYP-D3(BJ) and R = 3.29 Å for M062X). The minimum position only slightly grows (0.02 Å at all levels of theory: R = 3.37 Å for PBE-D3(BJ), R = 3.30 Å for B3LYP-D3(BJ) and R = 3.31 Å for M062X) when relaxing the C_37_H_16_ structure, and, as expected, the corresponding binding energies are more attractive from 30 to 70 meV. This might well be due to the increased interaction with the hydrogen atoms at the edges, an artificial effect which would be removed with the use of larger graphene prototypes. On the other hand, experimental adsorption energies and enthalpies on graphene are often found to be slightly more negative than the theoretical prediction, because graphene monolayers might present a wrinkled surface [40], where groove regions work as high-affinity sites enhancing the adsorption.

The effect of the use of larger, more realistic PAHs to model graphene sheets, such as C_54_H_18_ (circumcoronene), C_96_H_24_ (circumcircumcoronene) and C_150_H_30_ (Figure 5), was examined by comparing optimized interaction energies calculated at the previously validated best-performing PBE-D3(BJ) level of theory, which was therefore chosen for all subsequent calculations. The corresponding structures (in Cartesian coordinates) can be found in the Supporting Materials. Full optimizations, allowing both IMeAuCl and support free to relax (except for the largest C_150_H_30_ prototype), as well as partial optimizations of the free IMeAuCl onto rigid graphene models, were carried out, obtaining the BSSE-corrected and -uncorrected E_int_ values, reported in Table 4.

As was to be expected, the impact of the PAH moiety relaxation decreases as its size increases, a sign that the surface curvature is not substantially modified by IMeAuCl adsorption; the more negative interaction energies found for relaxed smaller models are essentially due to the artificially larger interaction with peripheral atoms. The limited change of the graphene model geometry upon increasing their size is also visible in Figure 6, where the side and top views of atom-by-atom superimposed structures of the IMeAuCl-C_54_H_18_ and IMeAuCl-C_96_H_24_, complexes are shown. The interaction energy difference between one model and the next larger one in fact decreases with the prototype size: the difference between the largest C_150_H_30_ and the minimal C_37_H_16_ one amounts to 15% ca. Such differences are lower when relaxed supports are employed due to a compensation between the increased interaction surface and the larger interaction with the atoms at the edge of the support.

The effect of BSSE on the calculated energies is rather relevant and slightly grows with the prototype size, i.e., with the presence of more spatially extended basis functions for only one monomer.

It can be interesting to compare the interaction energies for the Gold(I) monocarbene complex with those recently obtained for cisplatin (CP) on the same or very similar graphene models [33]. We, therefore, repeated the calculations at the PBE-D3(BJ)/6-311+G(2d,2p)+SDD level, the same as Ref. [33], i.e., by eliminating diffuse functions on the hydrogen atoms (Table 5). This slightly smaller basis set only leads to a few meV differences with the larger calculations of Table 4. Interaction energies for CP and IMeAuCl adsorption on the same models are very similar and slightly more attractive for the latter when BSSE-corrected energies are considered. This is mostly because the main source of interaction comes from the dispersion contribution, which, given the comparable size of the metal complexes, is expected to be similar, whereas no covalent or chemical type of interaction seems to come into play. In detail, the equilibrium distance between the metal complex and graphene is slightly shorter for Gold(I) monocarbene, and the dispersion and electrostatic attraction contribution to total interaction energies are more negative. However, the exchange repulsion is correspondingly larger, leading to overall very similar interaction energies.

Considering that rigid prototypes, particularly the largest ones, provide a good representation of the geometry and interactions occurring in a graphene sheet, these models have been used to compute adsorption energy (ΔE_ads_), enthalpy (ΔH_ads_) and free energy (ΔG_ads_) (Table 6), i.e., with respect to the relaxed monomers at infinite separation distance. For the largest IMeAuCl–C_150_H_30_ complex, for which only the rigid support interaction was considered, the conformation penalty, thermal corrections and entropy values were taken from the IMeAuCl–C_96_H_24_ model. Note that, for the three IMeAuCl–C_37_H_16_, IMeAuCl-C_54_H_18_ and IMeAuCl-C_96_H_24_ models, the differences in the conformational penalty (less than 3 meV) and in thermal corrections (less than 0.7 meV) are in fact extremely small and therefore will not have a large impact when transposed on larger models. ΔS_ads_ values are also very small and, as could be expected, grow with the size of the prototype: ΔS_ads_(IMeAuCl-C_37_H_16_) = 1.53 meV/K; ΔS_ads_(IMeAuCl-C_54_H_18_) = 1.55 meV/K; ΔS_ads_(IMeAuCl-C_96_H_24_) = 1.64 meV/K. The adsorption process is found to be very exothermic (−18.4 kcal/mol) and very exergonic (−7.20 kcal/mol), notwithstanding the unfavorable entropic contribution due to the loss of one translational degree of freedom upon adsorption. A comparison with the corresponding cisplatin adsorption values [33] for the slightly smaller CP–C_32_H_14_ complex at the PBE-D3(BJ)/6-311+G(2d,2p)+SDD level of theory is also reported in the same table and shows that the magnitude of the adsorption enthalpy and free energy is comparable. These favorable values indicate that graphene and its derivatives can be considered as promising nanomaterials for the load and delivery of gold(I)-carbene drugs. Indeed, these properties can be further enhanced or modulated by functionalizing graphene (e.g., by using coating or graphene oxides). Graphene oxide, alternating planar graphene-like and oxidized regions functionalized by epoxy and hydroxy groups, might further improve and modulate the adsorption properties of IMeAuCl. GO layers keep the same high surface-to-volume ratio and better biocompatibility than graphene, with the electrostatic and hydrogen bonding components of the interactions being enhanced. One can therefore take advantage of the solvent polarizability, of the cell pH, and generally of the environmental conditions to modify the GO adsorption/desorption ability. Such an effect has indeed already been observed experimentally [32] for the pH-dependent cisplatin load on GO; at basic pH values, the cisplatin charged on GO is maximum, whereas in an acidic environment (such as that occurring in cancer cells) cisplatin is released, partly due to the more favorable interactions of H_3_O^+^ groups with epoxy and hydroxy groups. Similar effects are expected to come into play for the adsorption of IMeAuCl on GO. Work in this direction is in progress.

Graphene coating with biocompatible materials is another way to take advantage of the graphene structure, endowing it with additional specific chemical, physical and medicinal properties, and it is very often found in applications [23,25] since bioactive compounds on the graphene surface can improve the efficiency of the loaded drug and increase its effectiveness. Molecules and polymers, such as doxorubicin, curcumin, chitosan and peptides, are often the chosen materials for coating when the delivery of anticancer compounds is involved. The modification of the IMeAuCl binding energy on such supports will obviously depend on the material selected for coating, and its exact determination would require a completely new set of calculations. However, based on what was found in the present investigation, one might suppose that the binding energy for the IMeAuCl molecules on such adsorbents will be similar to that found in the present work for materials with low polarity and/or with delocalized π electrons since dispersion interactions will again be the main responsible for adsorption.

## 3. Computational Methods

Benchmark potential energies for the interaction between IMeAuCl and the smallest considered graphene prototype (C_36_H_17_) were obtained at the density functional theory-symmetry adapted perturbation theory (DFT-SAPT) [41,42] and at the “coupled” second-order Møller-Plesset perturbation theory (MP2C) [43] level of theory by exploiting the Molpro2012.1 computational code [44]. In the case of DFT-SAPT, the PBE0 hybrid functional [45] was used to approximate the exchange–correlation functional, and the δ(HF) contribution [46], obtained at the Hartree–Fock (HF) level of theory and mostly accounting for the induction and exchange–induction effects higher than second-order, was also added to the total interaction energies. The aug-cc-pVDZ-PP [47] basis set was used for the Au atom together with the aug-cc-pVDZ [48] set for the remaining atoms. The complete basis set (CBS) limit for the DFT-SAPT energies is estimated by using the extrapolation scheme of Ref. [38], which consists of simply scaling the computed dispersion term by a factor of 1.1931.

As for the MP2C computations, which have been proven to provide accurate results for the interaction between aromatic species [49], the same basis sets as described above for the DFT-SAPT computations were used, and the obtained interaction energies were corrected for the basis set superposition error (BSSE) by using the Boys and Bernardi’s counterpoise technique [50]. In order to obtain the CBS limit for the MP2C energies the two-point correlation energy extrapolation scheme of Halkier et al. [51,52] was employed, which also needs the HF and MP2 interaction energies obtained with the contiguous aug-cc-pVDZ-PP/aug-cc-pVDZ and aug-cc-pVTZ-PP/aug-cc-pVTZ basis sets.

Corresponding interaction energies were estimated also at the DFT level using three distinct approaches, namely, PBE [53], B3LYP [54] and M062X [55], with the aim of validating them through the comparison with the benchmark DFT-SAPT and MP2C results. All DFT interaction energies were also corrected for the BSSE, and both PBE and B3LYP energies include the D3(BJ) [56] dispersion contribution correction. In these DFT calculations, the Au atom was described by the Stuttgart–Dresden pseudopotential (SDD) [57], whereas, for the rest of the atoms, the 6-311++G(2d,2p) [58] basis set was used.

Geometry optimization calculations to find the most stable structures and their energies were also carried out by exploiting the best performing DFT level (PBE-D3(BJ)), which was previously proven to be reliable for the description of the adsorption of a cisplatin (CP) molecule on finite graphene prototypes [34]. In particular, the abovementioned minima structures are those for the isolated IMeAuCl and for complexes involving the drug adsorbed on graphene prototypes of increasing size, namely, C_36_H_17_, C_54_H_18_, C_96_H_24_ and C_150_H_30_.

We followed the principles in Refs. [33,34] to produce an estimate of the related thermodynamic parameters such as the adsorption enthalpy (ΔH_ads_) and free energy (ΔG_ads_) of the clusters in the gas phase at 298 K and 1 atm. To obtain those parameters, the needed frequency calculations were performed by freezing the graphene prototype (GP) support while allowing the internal coordinates of the adsorbed drug to relax, and the rigid-rotor and harmonic oscillator approximations were also assumed.

All DFT computations were performed by using the Gaussian 09 [59] code.

## 4. Conclusions

We computationally investigated the interaction energy and thermodynamics of the adsorption of a Gold(I) monocarbene species, IMeAuCl, with relevant anticancer properties, onto a graphene sheet modeled by using prototypes of growing size, in view of a possible use of graphene-based nanostructures for drug delivery and the administration of Gold(I) N-heterocyclic carbenes, which are finding increasing applications in medicine.

After a benchmark of the best computational method in terms of accuracy and computational feasibility, we evaluated at the chosen PBE-D3(BJ)/6-311++G(2d,2p) level the interaction enthalpy and free energy for a sufficiently large PAH, C_150_H_30_, indicating a very favorable adsorption process.

The use of the accurate DFT-SAPT methods has also allowed us to characterize the physical components of the interaction, which provide useful information on how to functionalize or decorate the graphene surface to modulate the load/release capabilities of the drug carrier. In the present case, the favorable interaction energy and adsorption enthalpy mainly arise from the dispersion and, to a lesser extent, the electrostatic contributions. The latter component could be improved by the introduction of polar groups in graphene, as is the case for graphene oxide platelets or quantum dots. The adsorption of Gold(I) carbene molecules on graphene oxide and on functionalized graphene layers are currently under investigation and will be the subject of a subsequent publication.

## Figures and Tables

**Figure 1 molecules-28-03941-f001:**
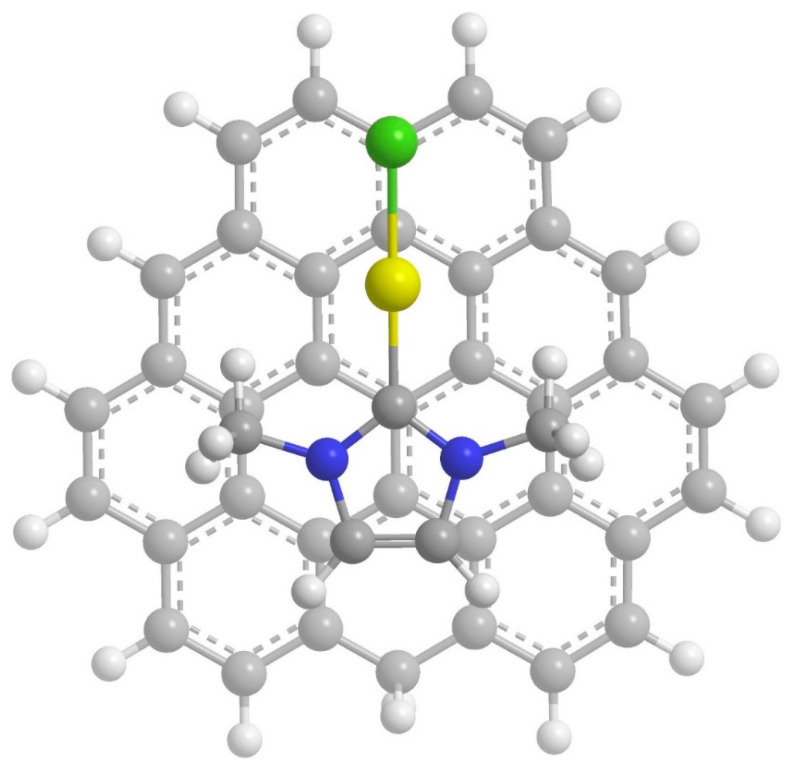
Minimum energy geometry for the interaction of IMeAuCl onto C_36_H_17_, the smallest investigated prototype.

**Figure 2 molecules-28-03941-f002:**
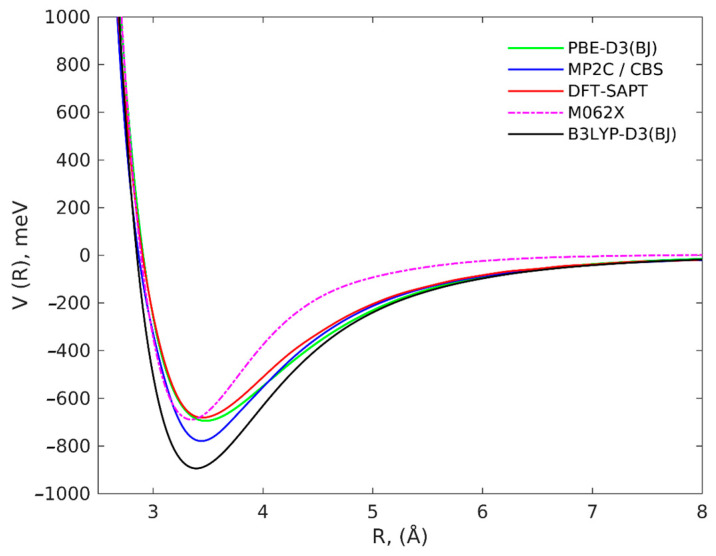
Interaction energy profiles for the parallel approach of IMeAuCl towards the C_37_H_16_ prototype, obtained at the investigated levels of theory: the benchmark MP2C/CBS and DFT-SAPT/CBS and the three functionals PBE-D3(BJ), M062X and B3LYP-D3(BJ), with the 6311++G(2d,2p) basis set.

**Figure 3 molecules-28-03941-f003:**
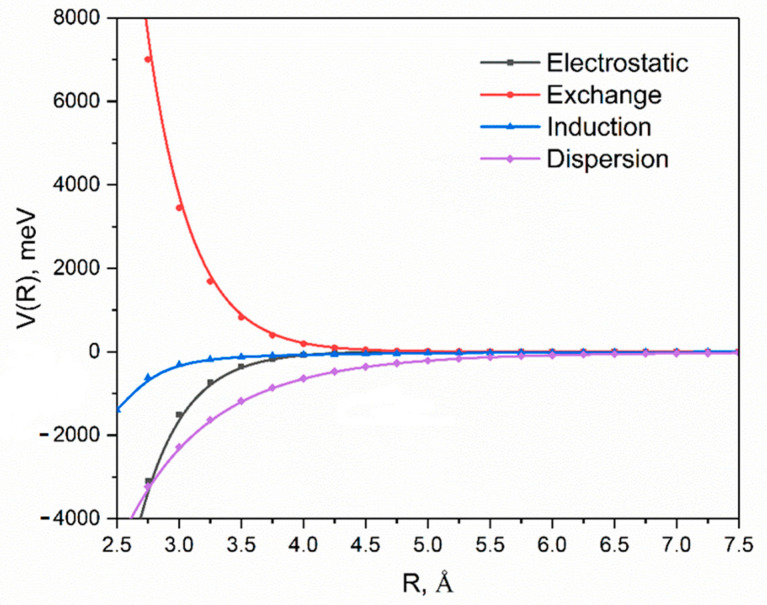
DFT-SAPT contributions (electrostatic, exchange repulsion, induction, dispersion) to the total interaction energy (red line in Figure 2) as a function of the intermolecular distance. Note that the dispersion contribution has been scaled by a factor of 1.1931 to obtain the CBS limit according to the scheme of Ref. [38].

**Figure 4 molecules-28-03941-f004:**
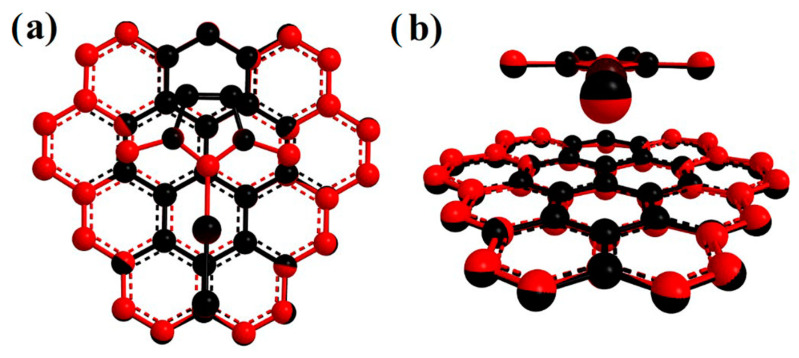
Atom-by-atom superimposition of the fully relaxed (red) over the rigid support structure (black) for the IMeAuCl-C_37_H_16_ geometries calculated at PBE-D3(BJ)/6-311++G(2d,2p)+SDD level of theory: (**a**) top view; (**b**) side view.

**Figure 5 molecules-28-03941-f005:**
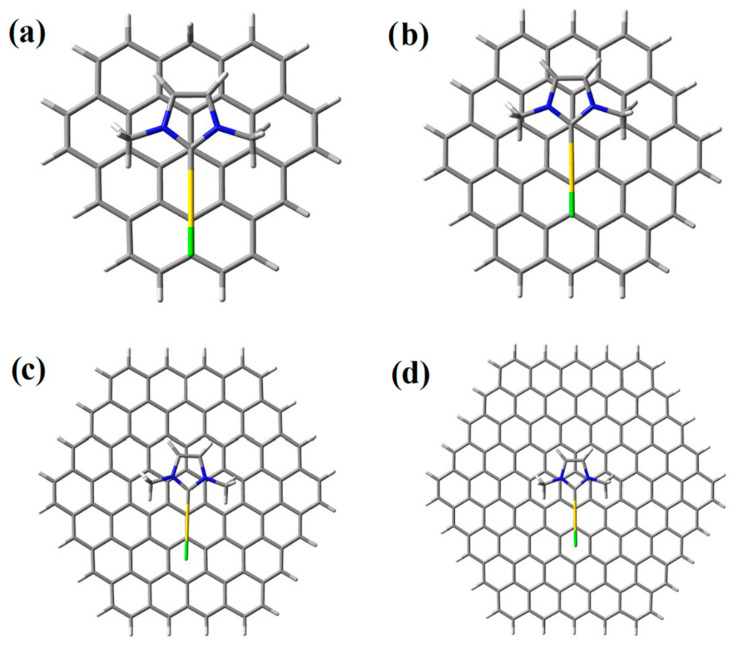
Adsorption geometries of IMeAuCl over increasingly larger graphene models: (**a**) C_37_H_16_, (**b**) C_54_H_18_, (**c**) C_96_H_24_ and (**d**) C_150_H_30_ complexes.

**Figure 6 molecules-28-03941-f006:**
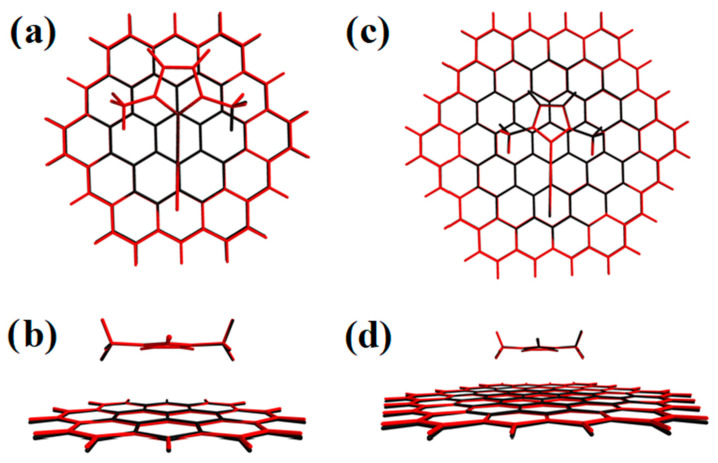
Top (**a**,**c**) and side (**b**,**d**) views of IMeAuCl-Circumcoronene (C_54_H_18_) and IMeAuCl-Circumcircumcoronene (C_96_H_24_) structures for the fully relaxed (red) over the rigid support structure (black).

**Table 1 molecules-28-03941-t001:** DFT-SAPT/CBS contributions to the minimum interaction energy between IMeAuCl and C_37_H_16_ in their rigid geometries.

	Energy (meV)
Electrostatic	−429.3
Exchange	1002.4
Induction	−126.9
Dispersion ^a^	−1314.7
δ(HF)	−46.6
Total	−702.9

^a^ E_d_ value was multiplied by 1.1931 for the extrapolation to the CBS limit [38].

**Table 2 molecules-28-03941-t002:** Selected bond lengths (Å) and angles (deg) of isolated and C_37_H_16_ adsorbed IMeAuCl calculated at the PBE-D3(BJ), B3LYP-D3(BJ) and M062X levels of theory with the 6-311++G(2d,2p) + SDD basis set. Experimental X-rays values [39] are also reported for comparison.

	Crystal Structure [39]	Monomer			Complex		
		PBE-D3(BJ)	B3LYP-D3(BJ)	M062X	PBE-D3(BJ)	B3LYP-D3(BJ)	M062X
Au–Cl	2.2882	2.2994	2.3145	2.3297	2.3089	2.3250	2.3506
Au–C1	1.9787	1.9864	2.0066	2.0047	1.9879	2.0067	2.0076
C1–N1	1.35	1.3665	1.3560	1.3503	1.3657	1.3551	1.3488
C1–N2	1.347	1.3665	1.3560	1.3503	1.3657	1.3551	1.3488
C2–N1	1.36	1.3853	1.3831	1.3785	1.3853	1.3833	1.3772
C3–N2	1.39	1.3853	1.3831	1.3785	1.3853	1.3833	1.3771
C1-Au-Cl	178.8	179.9	180.0	179.9	179.0	178.6	179.9
N1-C1-N2	105.8	104.4	104.7	104.7	104.5	104.8	104.8

**Table 3 molecules-28-03941-t003:** BSSE-corrected interaction energies E_int_ between IMeAuCl and C_37_H_16_ The energy values were calculated as single points on the rigid support and fully relaxed geometries of the minimum IMeAuCl-C_37_H_16_ structure. DFT-SAPT and MP2C calculations were performed on the optimized PBE-D3(BJ)/6-311++G(2d,2p) geometry.

C_37_H_16_ Geometry	PBE-D3(BJ)/ 6-311++G(2d,2p)	DFT-SAPT/CBS	MP2C/ CBS	B3LYP-D3(BJ)/ 6-311++G(2d,2p)	M062X/ 6-311++G(2d,2p)
Rigid	−731.6	−702.9	−804.54	−955.3	−786.6
Relaxed	−783.2	−734.7	−855.71	−1023.3	−852.5

**Table 4 molecules-28-03941-t004:** BSSE-corrected and -uncorrected interaction energies at PBE-D3(BJ)/6-311++G(2d,2p)+SSD level of theory for graphene models of increasing size. The values are obtained by optimizing IMeAuCl on rigid and relaxed graphene prototypes.

	R (Å)	BSSE Corrected (meV)	BSSE Uncorrected (meV)
IMeAuCl-C_37_H_16_ (rigid/relaxed)	3.35/3.37	−731.6/−783.2	−831.8/885.1
IMeAuCl-C_54_H_18_ (rigid/relaxed)	3.34/3.37	−805.1/−829.1	−918.1/−946.6
IMeAuCl-C_96_H_24_ (rigid/relaxed)	3.31/3.38	−836.9/−853.8	−951.8/−976.9
IMeAuCl-C_150_H_30_ (rigid)	3.30	−861.3	−988.5

**Table 5 molecules-28-03941-t005:** BSSE-corrected and -uncorrected interaction energies for the adsorption of IMeAuCl and Cisplatin (CP) (the latter reported in italics) on rigid graphene molecular prototypes of increasing size calculated at PBE-D3(BJ)/6-311+(2d,2p)+SDD level of theory.

	BSSE-Corrected (meV)	BSSE-Uncorrected (meV)
IMeAuCl-C_37_H_16_	−734.6	−834.6
IMeAuCl-C_54_H_18_/*CP*-*C_54_H_18_* [34]	−805.3/−*794.1*	−917.7/−*939.4*
IMeAuCl-C_96_H_24_/*CP*-*C_96_H_24_* [34]	−840.0/−*818.8*	−960.8/−*963.7*
IMeAuCl-C_150_H_30_	−862.0	−982.2

**Table 6 molecules-28-03941-t006:** BSSE-corrected (uncorrected) interaction (E_int_) and adsorption energies (ΔE_ads_), enthalpy (ΔH_ads_) and free energies (ΔG_ads_) for the adsorption of IMeAuCl on rigid graphene molecular prototypes at PBE-D3(BJ)/6-311++G(2d,2p) level of theory. Data for the interaction between cisplatin and C_32_H_14_ are taken from ref. [33]. All energies are in meV.

	E_int_	ΔE_ads_	ΔH_ads_	ΔG_ads_
IMeAuCl-C_37_H_16_	−731.6 (−831.8)	−719.9 (−820.1)	−664.4 (−764.6)	−209.7 (−309.9)
IMeAuCl-C_54_H_18_	−805.1 (−918.1)	−795.0 (−908.0)	−739.1 (−852.1)	−276.5 (−389.5)
IMeAuCl-C_96_H_24_	−836.9 (−951.8)	−829.5 (−944.4)	−774.6 (−889.5)	−286.8 (−401.7)
IMeAuCl-C_150_H_30_	−862.0 (−982.2)	−854.6 (−974.8) ^a^	−799.7 (−919.9) ^a^	−311.8 (−432.0) ^a^
CP-C_32_H_14_ [33]	−744.7	(−839.3)	(−785.8)	(−324.7)

^a^ value estimated by using the conformational penalty, thermal corrections and ΔS_ads_ calculated for IMeAuCl-C_96_H_24_.

## Data Availability

The data presented in this study are available in the article and in the Appendix A.

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
