# Peer review of "Graphene as Nanocarrier for Gold(I)-Monocarbene Complexes: Strength and Nature of Physisorption"

_molecules, 2023, doi:10.3390/molecules28093941_

Round 1

Reviewer 1 Report

This is an interesting paper regarding the possible use of graphene for use as an agent for the delivery of Gold metal complexes for cancer treatment. I have no comments on the methods described in the paper but rather in how the paper is contextualized in potential applications which should be included in the discussion part of the paper.

1) The paper talks about the adsorption energies under different models for IMeAuCl idealizeds graphene. I would be interesting on seeing how this energy will translate to the desorption necessary for delivery of this drug once it penetrates cells.

2) Give a context for the observed energy values with other potential adsorbents from the literature. This can be other drug molecules or other agents with similar proposed binding modes.

3) Graphene oxides may be a more readily obtained nanoagent for the application as agents for drug delivery. Can you include discussion on how it is expected the presence of oxygen species would have on these binding modes?

4) Other agents that may be interesting would be the formation of composite species with other materials. How would that form of modification alter the observed binding energies? 

Reviewer 2 Report

Dear Editor

Revision Report for the Article "Graphene as nanocarrier for gold(I)-Monocarbene complexes: strength and nature of physisorption"

I have investigated the manuscript in detail and my comments are listed below;

The submitted document discusses the interaction properties between adsorbate and the adsorbent. The authors theoretically evaluated these properties of interaction. The authors used several basic and functional methods, but they did not clearly state which method was the most adequate. A clear justification and interpretation must be presented during the review. I suggest to the authors to justify the nature of the physisorption.

Upon careful evaluation of the document and based on the similarity report, it was found that the similarity rate is above 39%. Therefore, we recommend a major revision of the article.

We request that the authors thoroughly review the document and correct all parts of the text identified as similar to other sources. The authors must also provide accurate and appropriate references for all information borrowed from other sources. We also suggest that the authors use plagiarism detection software to ensure that the similarity rate is well below the acceptable threshold.

Furthermore, we recommend that the authors enhance the clarity and conciseness of the methodology used to evaluate the interaction properties of gold(I)-Monocarbene complexes. It is important that the methodology be clearly explained and well-documented to allow for reproducibility of results.

In conclusion, despite the potential of the document to make a significant contribution to the field of evaluating the interaction properties of gold(I)-Monocarbene complexes, a major revision is necessary to ensure the academic integrity of the article. We look forward to receiving the revised version of the article.

Acceptable

Round 2

Reviewer 2 Report

Thank you for allowing me to review the manuscript entitled” Graphene as nanocarrier for gold(I)-Monocarbene complexes: strength and nature of physisorption”.

The revised submission answered all queries and concerns, with all the points suggested by referees highlighted. From my perspective, the paper has dramatically improved.

I believe that a manuscript in this style can be accepted for publication in Molecules Journal in its current form.

Acceptable